



# Tethered balloon-borne observations of thermal-infrared irradiance and cooling rate profiles in the Arctic atmospheric boundary layer

Michael Lonardi[1], Elisa F. Akansu[2], André Ehrlich[1], Mauro Mazzola[3], Christian Pilz[2], Matthew D. Shupe[4,5], Holger Siebert[2], and Manfred Wendisch[1]

[1]Leipzig Institute for Meteorology (LIM), Leipzig University, Leipzig, Germany
[2]Leibniz Institute for Tropospheric Research (TROPOS), Leipzig, Germany
[3]Institute of Polar Sciences (ISP), Italian National Research Council (CNR), Bologna, Italy
[4]Cooperative Institute for Research in Environmental Sciences (CIRES), University of Colorado, Boulder, Colorado, USA
[5]Physical Science Laboratory, National Oceanic and Atmospheric Administration (NOAA), Boulder, Colorado, USA

**Correspondence:** Michael Lonardi (michael.lonardi@uni-leipzig.de)

**Abstract.** Clouds play an important role in controlling the radiative energy budget of the Arctic atmospheric boundary layer. To quantify their impact on diabatic heating or cooling of the atmosphere and of the surface, vertical profile observations of thermal-infrared irradiances were collected using a tethered balloon. We present 70 profiles of thermal-infrared radiative quantities measured in summer 2020 at the Multidisciplinary drifting Observatory for the Study of Arctic Climate (MOSAiC) expedition, and in autumn 2021 and spring 2022 in Ny-Ålesund, Svalbard. Measurements are classified into four groups: cloudless, low-level liquid-bearing cloud, elevated liquid-bearing cloud, and elevated ice cloud. Cloudless cases display a radiative cooling rate of about -2 K day$^{-1}$. Observed low-level liquid-bearing clouds are characterized by a radiative cooling up to -80 K day$^{-1}$ in a shallow layer at cloud top. Radiative transfer simulations are performed to quantify the sensitivity of radiative cooling rates to cloud microphysical properties. In particular, cloud top cooling has a strong response to variation of the liquid water path, especially in optically thin clouds, while for optically thick clouds the cloud droplet number concentration has an increased relative importance. Two case studies with a changing cloud cover are presented to investigate the temporal evolution of radiation profiles during the transitions between (a) cloudy to cloudless and (b) low-level to elevated clouds. Additional radiative transfer simulations are used to reproduce the observed scenarios and to showcase the radiative impacts of elevated liquid and ice clouds, demonstrating the increased radiative significance of the liquid clouds.

## 1 Introduction

The Arctic climate system is undergoing dramatic changes that exceed the average global warming signal. The enhanced warming in this region is a result of different feedback mechanisms known as Arctic amplification (Serreze and Barry, 2011; Wendisch et al., 2023). In this framework, the role of clouds and their realistic representation in models is still uncertain, mainly because of their frequent occurrence (Shupe et al., 2011; Mioche et al., 2015) and the complexity of cloud processes (Curry, 1986; Curry et al., 1996; Morrison et al., 2012; Wendisch et al., 2019). In particular, clouds exert a strong control on the Arctic surface energy balance (Intrieri et al., 2002; Becker et al., 2023). They have a cooling impact by reducing the



incoming solar radiation, and a warming effect by emitting thermal-infrared (TIR) radiation (Shupe and Intrieri, 2004). The total (solar plus TIR) cloud radiative effect is strongly variable and depends on the solar zenith angle, thermodynamic and microphysical properties of the cloud, and the surface albedo and temperature (Shupe and Intrieri, 2004). Cloud radiative

properties are strongly affected by the thermodynamic phase of the cloud (Sun and Shine, 1994; Ehrlich et al., 2009), with liquid-bearing clouds having a stronger radiative effect than ice-only clouds (Shupe and Intrieri, 2004; Turner et al., 2018). Liquid-bearing clouds can be purely liquid or can contain also an ice component (mixed-phase clouds). The liquid water path (LWP) largely controls the attenuation of incoming solar radiation and the emission of TIR radiation, with optically thick clouds (LWP $\geq 30\,\mathrm{g\,m^{-2}}$) emitting nearly as black bodies (Shupe and Intrieri, 2004; Turner et al., 2018). Williams and Igel

(2021) and Morrison et al. (2008) showed that droplet number concentration ($N_\mathrm{d}$) impacts the optical properties of clouds. The interaction of clouds with surface albedo and temperature has been studied over the Arctic sea ice (Curry and Ebert, 1992; Walsh and Chapman, 1998; Intrieri et al., 2002; Walden et al., 2017), open ocean (Wendisch et al., 2022; Becker et al., 2023), and land (Dong et al., 2010; Miller et al., 2015; Ebell et al., 2020).

While most of the literature discusses the cloud radiative effect at the surface, the atmospheric boundary layer (ABL) is

also affected by local radiative energy sources and sinks at different altitudes (Curry, 1986; Yamamoto et al., 1995; Asano et al., 2004; Philipona et al., 2020). In particular, Turner et al. (2018) showed that the vertical structure of irradiance and radiative cooling rate profiles depends on the atmospheric state. In a cloudless atmosphere, the emission of TIR radiation from the surface results in cooling that promotes the formation of a surface-based temperature inversion. In liquid-bearing clouds, cloud-top radiative cooling contributes to strong stratification just above the cloud layer, thus de-coupling the cloud,

and consequently the ABL, from the overlying atmosphere. In turn, the vertical motion of sinking colder air parcels induces mixing in and below the cloud, which is important for cloud maintenance (e.g., Morrison et al., 2012). Entrainment of moisture (Shupe et al., 2013; Solomon et al., 2014; Egerer et al., 2021) and cloud condensation nuclei from above the cloud can also prolong the lifetime of the cloud. Conversely, the reduction of radiative cooling at cloud top due to the shading effects of a second cloud at higher altitudes dampens the vertical mixing, which may shorten the cloud lifetime (Shupe et al., 2013; Turner

et al., 2018; Chechin et al., 2022).

While there are a variety of cloud structures in the Arctic, it is typically not possible to distinguish their vertical properties using surface TIR measurements alone. Furthermore, cloud scenes are not stationary and transitions may be driven locally or by larger-scale drivers, like advection from different regions or at different altitudes. Transitions may occur at a fixed location when two air masses are exchanged, or along air mass trajectories when air masses are transformed (Pithan et al., 2018).

Radiosondes provide in situ measurements of the thermodynamic state over the vertical axis, but the interaction among the atmospheric variables is not instantaneous, thus isolated radiosonde profiles are not sufficient to fully depict the processes involved. Profile observations of radiative properties in the cloudy ABL with high vertical and temporal resolution are suited to close this measurement gap, but they become particularly challenging in the harsh Arctic environmental conditions. Nevertheless, in recent years a significant amount of data were collected with the help of aircraft (Ehrlich et al., 2019; Wendisch

et al., 2019, 2022; Becker et al., 2023), free-flying balloons (Philipona et al., 2020), and tethered balloons (Lawson et al., 2011; Sikand et al., 2013; Dexheimer et al., 2019; Becker et al., 2020; Inoue et al., 2021). In particular, combined measurements



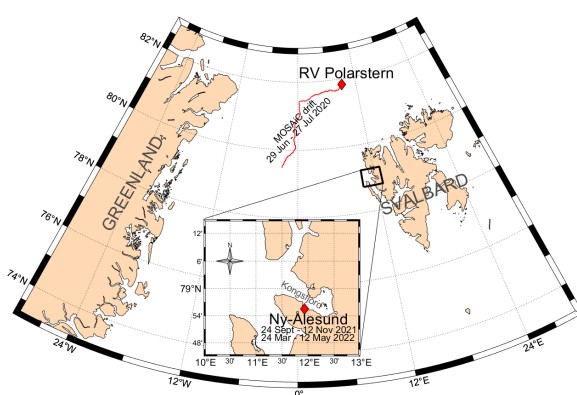

**Figure 1.** Drift track of the MOSAiC ice camp in summer 2020 (red line), and geographical location of Ny-Ålesund. The box displays the location of the measurement site with respect to the fjord.

of thermodynamic properties (Egerer et al., 2021), broadband irradiances (Lonardi et al., 2022), turbulence (Egerer et al., 2019, 2023), and aerosol particle properties (Pilz et al., 2022) were obtained with the tethered balloon-borne system BELUGA (Balloon-bornE moduLar Utility for profilinG the lower Atmosphere; Egerer et al., 2019).

In this study, BELUGA profile observations of TIR irradiances and derived parameters (net irradiance, radiative cooling rate) are analyzed to characterize the effect of atmospheric state and cloud properties on net irradiance and cooling rate profiles in the Arctic. Measurements were obtained in summer 2020 during the Multidisciplinary drifting Observatory for the Study of Arctic Climate (MOSAiC) expedition (Shupe et al., 2022), and during autumn 2021 and spring 2022 in Ny-Ålesund, Svalbard. In Sect. 2 the instrumental setup and the methods used to derive vertically resolved profiles of net irradiance and radiative

cooling rates are described. The data of all measurement periods are statistically analyzed in Sect. 3 characterizing different atmospheric states. The temporal evolution of radiation profiles during the transition from one state to another is characterized by sequences of balloon profile observations and discussed in Sect. 4. In Sect. 5, radiative transfer simulations are used to quantify the sensitivity of radiative cooling rates to cloud microphysics properties of elevated clouds.

## 2    Instruments and methods

Measurements in the Arctic lower troposphere were performed at a drifting ice floe during MOSAiC (Shupe et al., 2022) between 29 June–27 July 2020 (hereafter referred to as "summer"), and at the permanent joint German–French AWIPEV (Alfred Wegener Institute for Polar and Marine Research and the French Polar Institute Paul Emile Victor) research base in Ny-Ålesund (Svalbard) between 24 September–12 November 2021 and between 24 March–12 May 2022 ("autumn" and "spring", respectively). Figure 1 displays the geographical position of the measurement sites. Although the MOSAiC measurements

were obtained while drifting southwards through the Fram Strait, which is almost the same latitude compared to Ny-Ålesund



(79° N), the two sites represent different environmental conditions. The MOSAiC camp was placed on an ice floe covered with melting snow, surrounded by an increasing fraction of melt ponds and open water. Shupe et al. (2022, Fig. 3) visually displays the balloon site in this environment. The Ny-Ålesund ground site was located next to a small settlement close to the generally ice-free Kongsfjord. The fjord is oriented northwest-southeast and is surrounded by plateaus and mountains of about
500-1000 m altitude. Their presence affects the local wind circulation and the advection of air masses (Maturilli and Kayser, 2017; Gierens et al., 2020; Schön et al., 2022).

At both sites, the tethered balloon setup was complemented by a comprehensive observation network suite with near-surface radiation measurements, surface-based remote sensing, and radiosoundings (daily at Ny-Ålesund, 6-hourly at MOSAiC). Shupe et al. (2022) display a detailed overview of the instrumentation deployed at the MOSAiC site, while the Ny-Ålesund measure-
ment setup includes tower measurements (Mazzola et al., 2016), ground-based remote sensing (Nomokonova et al., 2019), and radiosoundings (Maturilli et al., 2013).

Tethered balloon measurements were conducted using BELUGA (Egerer et al., 2019) to profile the Arctic lower troposphere. The core measurement setup used during the three deployments is described by Lonardi et al. (2022) and Pilz et al. (in review). During MOSAiC the observations were obtained by continuously profiling the lower troposphere up to 1500 m with an ascent
rate of 0.5-1 m s$^{-1}$, while in Ny-Ålesund the profiles were occasionally stopped at a fixed height for 2 h to allow filter sampling of aerosol particles.

The broadband radiation package (BP) measured upward and downward TIR (4.5-42 µm) and solar (0.3-2.8 µm) broadband irradiances. Due to the absence of solar radiation in autumn, and payload limitations during the observations in spring 2022, BP was substituted by a lighter configuration measuring TIR irradiances only. For the analysis of vertical profiles of cooling
rates presented here, the solar component is of secondary importance (Turner et al., 2018), even in mid-summer (Lonardi et al., 2022), although it might become relevant when investigating the evaporation of cloud particles at cloud top or cloud-surface albedo interactions. Therefore, we analyze only the TIR irradiances.

Balloon-borne measurements of upward irradiance ($F^{\uparrow}$) and downward irradiance ($F^{\downarrow}$) were radiometrically calibrated using the methods introduced by Egerer et al. (2019); Lonardi et al. (2022); Pilz et al. (in review), and showed agreement with
permanent surface observations within the limits of their uncertainties (7 W m$^{-2}$). To quantify the radiative energy budget at flight altitude $z$, $F^{\uparrow}$ and $F^{\downarrow}$ were used to derive the net irradiance ($F_{\text{net}}$) defined by:

$$F_{\text{net}}(z) = F^{\downarrow}(z) - F^{\uparrow}(z), \tag{1}$$

The vertical divergence of $F_{\text{net}}$ in an atmospheric layer results in local temperature tendencies (cooling or warming). These temperature tendencies induced by radiation in a layer between two heights $z_{\text{bot}}$ and $z_{\text{top}}$ can be quantified by the radiative
temperature tendency rate ($\zeta$) using:

$$\zeta = \frac{1}{\rho \cdot c_{\text{p}}} \frac{F_{\text{net}}(z_{\text{top}}) - F_{\text{net}}(z_{\text{bot}})}{z_{\text{top}} - z_{\text{bot}}} \tag{2}$$

where $\rho$ is the density of the air and $c_{\text{p}}$ is the specific heat capacity at constant pressure. While they are referring to the same process, in this paper we will refer to positive values as "heating rates" and negative values as "cooling rates". The magnitude





of $\zeta$ depends on the layer thickness over which the convergence or divergence of radiative fluxes occurs. To be consistent, we
first interpolated the irradiance profiles to a common vertical grid with a $1\,\mathrm{m}$ resolution, then we calculated the $\zeta$ profiles using
a layer thickness of $10\,\mathrm{m}$. Such resolution allows for characterizing the temperature tendencies also in the narrow cloud top
region.

Measurements were compared to simulations by the plan-parallel radiative transfer model DISORT (Stamnes et al., 1988)
implemented in the libRadtran 2.0.3 software package (Emde et al., 2016). The gaseous absorption parameterization used
in the model was LOWTRAN (Ricchiazzi et al., 1998). Temperature and water vapor in the simulations were based on the
nearest-in-time radiosonde data complemented in the lower layers by BELUGA observations.

## 3 Radiative profiles in cloudless and cloudy conditions

### 3.1 Classification of atmospheric states

A total amount of 70 profile observations including BP were collected with BELUGA during the three campaigns. Each
balloon ascent was performed up to a maximum height of $1000$–$1500\,\mathrm{m}$ in a time span of about 30–45 minutes. 17 profiles
were performed in summer 2020 during MOSAiC, 36 profiles were obtained in Ny-Ålesund in autumn 2021, and 17 in spring
2022. The atmospheric variables measured during the three investigated periods cover a large fraction of the variability of the
Arctic lower troposphere, although cold, stable winter conditions over sea ice are missing.

To characterize the impact of clouds on radiative properties during different atmospheric states or transitions between them,
balloon-borne profiles were sorted based on the cloud properties retrieved from the Cloudnet products (Illingworth et al., 2007;
Griesche et al., 2020). If a liquid-bearing cloud (LWP $\geq 5\,\mathrm{g\,m^{-2}}$) was the only cloud layer present, and the balloon profiled
through cloud top, the case was defined as a "low-level liquid-bearing cloud". If cloud top was not reached, or if a second
cloud was present above, the case was an "elevated liquid-bearing cloud". If clouds contained ice and had little-to-no liquid
water (LWP $< 5\,\mathrm{g\,m^{-2}}$), the case was classified as an "elevated ice cloud". In this case, the additional specification on the cloud
height comes from the fact that no ice cloud was reached by the balloon measurements. Table 1 summarizes the categories and
the relative counts sorted per season.

The Arctic summer atmosphere over the pack ice is typically covered by fog and/or liquid-bearing low clouds (Curry et al.,
1996; Shupe, 2011; Tjernström et al., 2012; Brooks et al., 2017). Rinke et al. (2021) showed that summer 2020 was unusually
warm at the MOSAiC site, resulting in a lower share of ice-bearing clouds in favor of purely liquid-water clouds (e.g., Lonardi
et al., 2022). Within the BELUGA operations in summer 2020, low-level clouds frequently occurred alone (eight cases) or
were capped by an additional cloud layer (elevated liquid-bearing cloud, six cases), while only three cases were cloudless. A
study conducted by Nomokonova et al. (2019) between 2016 and 2017 showed that Ny-Ålesund is generally characterized by
the presence of clouds throughout the entire year (81%), indicating also that in autumn clouds are typically centered at about
$1\,\mathrm{km}$ height, close to the ceiling height for tethered balloon profiles. BELUGA data in autumn 2021 consistently showed a vast
presence of elevated clouds (13 cases) compared to low-level clouds (four cases), which were inhibited by the local topography.





**Table 1.** Prevailing atmospheric state during the measurement profiles performed in summer (MOSAiC) and during autumn and spring (Ny-Ålesund).

|  | Summer (MOSAiC) | Autumn (Ny-Ålesund) | Spring (Ny-Ålesund) | Total |
|---|---|---|---|---|
| **Cloudless** | 3 | 12 | 12 | 27 |
| **Low-level liquid-bearing clouds** | 8 | 4 | 1 | 13 |
| **Elevated liquid-bearing clouds** | 6 | 13 | 1 | 20 |
| **Elevated ice clouds** | - | 7 | 3 | 10 |
| **Total** | 17 | 36 | 17 | 70 |

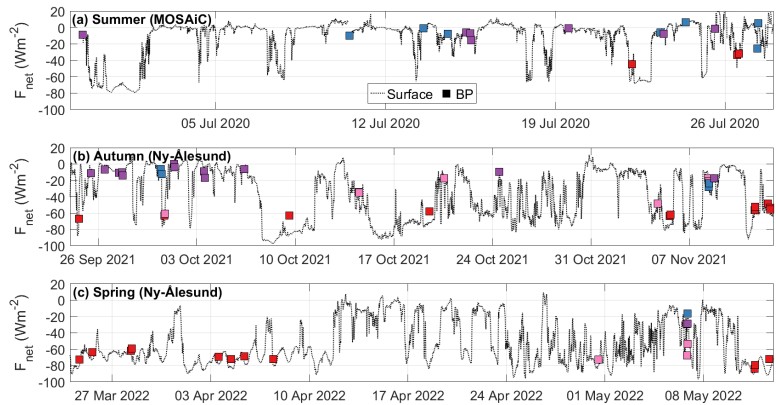

**Figure 2.** TIR net irradiances in summer (a), autumn (b), and spring (c). Time series (dotted lines) show 2 m observations by the ASFS at the MOSAiC ice-floe, and 33 m observations by the CCT in Ny-Ålesund. Balloon in situ measurements at 33 m (squares) during ascents are color-coded to indicate the atmospheric state: cloudless (red), low-level liquid-bearing clouds (blue), elevated liquid-bearing clouds (purple), and elevated ice clouds (pink). Note that the three time periods have different lengths.

Several cloudless scenarios (12 cases) occurred in autumn, and they were the dominant state (12 cases) observed in spring, partially in agreement with an increased share of cloudless scenarios in spring reported by Nomokonova et al. (2019).

Figure 2 displays the time series of TIR net irradiances measured at the Atmospheric Surface Flux Station (ASFS, 2 m, Cox et al., 2023) and at the Climate Change Tower (CCT, 33 m) during the deployment periods at MOSAiC ice floe and in Ny-Ålesund, respectively. Tethered balloon profiles are characterized by BP measurements close to the surface (33 m, at both sites), and are color-coded following the atmospheric state. This height was selected to neglect possible surface disturbances caused by the operations at the balloon site. Due to logistical and meteorological limitations in the operation of BELUGA (surface



**Figure 3.** Frequency of TIR net irradiances near the surface (a-c) and at maximum height (d-e) during the balloon deployments in summer, autumn, and spring. Each balloon profile was sampled only once per level (once at 33 m and once at the maximum height level), and the time-corresponding data point of the time series was used to obtain the subset of the surface distribution. Surface distributions were derived from the time series from surface-based systems (ASFS for MOSAiC, CCT for Ny-Ålesund) displayed in Figure 2.

wind needs to be below $5\,\mathrm{m\,s^{-1}}$), observations are not equally distributed over the campaign period. Except for the first days of July 2020, liquid-bearing clouds were almost continuously observed by Cloudnet at the MOSAiC ice floe site, as indicated

by small values of TIR net irradiances (Fig. 2a). In Ny-Ålesund, the first weeks of autumn 2021 were characterized by liquid-bearing elevated or multilayer clouds, while later on ice-only clouds and cloudless regimes prevailed (Fig. 2b). Early spring 2022 was typically cloudless, with clouds frequently appearing only after mid-April (Fig. 2c). In general, frontal passages, characterized by cloud cover, were not sampled with BELUGA due to the strong winds associated with the synoptic activity.

To quantify the representativeness of the BELUGA profiles to the entire campaign conditions, the frequency distributions

of near-surface TIR net irradiances of the entire campaign periods and the BELUGA flights are displayed in Figure 3. Based on the distribution of near-surface observations in Figure 2, a threshold of $F_\mathrm{net} = -30\,\mathrm{W\,m^{-2}}$ is defined to distinguish between "cloudless" (less than $-30\,\mathrm{W\,m^{-2}}$) and "cloudy" (larger or equal than $-30\,\mathrm{W\,m^{-2}}$) cases. In cloudless conditions, a strongly negative TIR radiative budget is observed. In cloudy cases, the presence of one or multiple liquid-bearing clouds increases the $F^\downarrow$ resulting in net irradiances close to zero. During all three deployment periods, the bimodal distribution associated with

cloudless/cloudy conditions (Shupe and Intrieri, 2004; Stramler et al., 2011; Wendisch et al., 2019, 2022; Solomon et al., 2023) was present at the surface, although significant differences in the strength of the two modes are observed.

There is a general match between the distribution of irradiances for the BELUGA near-surface measurements (red bars in Fig. 3a-c) and the surface measurements at the same times (dashed bars in Fig. 3a-c). However, the time periods of the BELUGA observations are only partially representative of the overall distribution of conditions (black bars in Fig. 3a-c). Surface

measurements in summer displayed a scarcity of cloudless cases in favor of cloudy periods (Fig. 3a). BELUGA observation periods captured the cloudy cases of the period but largely missed the cloudless component due to intermittent sampling. Ground measurements in autumn and spring indicated a more evenly distributed frequency between the two modes, with the cloudy mode being still dominant during autumn (Fig. 3b) and the cloudless mode prevailing in spring (Fig. 3c). BELUGA profiles in autumn generally followed the seasonal distribution, while in spring the profiles were performed more frequently during

cloudless conditions. The differences in net irradiances in cloudless atmospheres in Ny-Ålesund in comparison to the summer are caused by the temperature difference among seasons. In cloudy cases, net irradiances at the surface were typically negative in all seasons. However, for some cases at MOSAiC, the TIR net irradiance was directed toward the surface, indicating the advection of clouds that were warmer than the melting ice surface (locked at $0\,°\mathrm{C}$).

The distribution of surface measurements did not vary significantly during the balloon ascents (dashed bars in Fig. 3d-f),

while the distribution observed at the top of BELUGA profiles (blue bars in Fig. 3d-e) was different due to the vertical variation of the cloud cover. The occurrence of strongly negative irradiances at maximum height indicates that the cloudy atmospheres



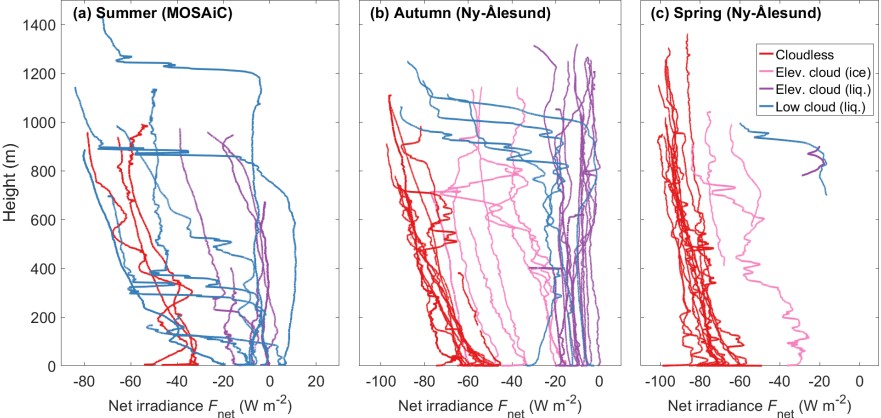

**Figure 4.** Observed profiles of net irradiance in (a) summer, (b) autumn, and (c) spring. The color coding indicates the classification of the atmospheric states: cloudless (red), low-level liquid-bearing clouds (blue), elevated liquid-bearing clouds (purple), and elevated ice clouds (pink). The calculated uncertainty of each data point is $\pm 7\,\mathrm{W\,m^{-2}}$.

profiled at MOSAiC mostly featured only a low-level cloud (Fig. 3d). Therefore, above the cloud, $F_{\mathrm{net}}$ has the values of a cloudless state. BELUGA observations at maximum height in autumn largely represent the same atmospheric state (cloudy vs cloudless, Fig. 3e), indicating that generally the cloud cover consisted of elevated clouds. However, a decrease in the number of cloudy cases in favor of cloudless ones indicates that some of the observations in cloudy scenarios reached above the cloud layer, suggesting that these cases featured a low-level cloud. The cloud case observed in spring (Fig. 3f) was characterized by an elevated ice cloud, therefore the net irradiance decreased with height, and at the top of the profile could be interpreted as a cloudless case.

### 3.2 Net irradiance profiles

Figure 4 summarizes all observations of $F_{\mathrm{net}}$ profiles obtained in the three campaign periods. The profiles were classified according to the atmospheric states introduced in Sect. 2. Cloudless profiles (red lines in Fig. 4) exhibit a steady decrease of $F_{\mathrm{net}}$ with height at a rate of about $-2.5\,\mathrm{W\,m^{-2}}$ over $100\,\mathrm{m}$ altitude in spring and $-3\,\mathrm{W\,m^{-2}}$ over $100\,\mathrm{m}$ altitude in autumn. Overall, the vertical gradient is a result of the temperature profiles, where colder higher layers emit less downward irradiance than the warmer low layers emit upward (following the Stefan–Boltzmann law). At low altitudes, the surface dominates due to its higher emissivity compared to the atmosphere, resulting in a decreasing $F_{\mathrm{net}}$ despite the presence of temperature inversions. $F_{\mathrm{net}}$ in the presence of elevated ice clouds (pink lines in Fig. 4) shows less negative values due to additional emission of downward irradiance by the cloud, which is located in atmospheric layers with cold temperatures but has a higher emissivity than the atmosphere alone. However, the vertically decrease of $F_{\mathrm{net}}$ has a rate on the order of $-2\,\mathrm{W\,m^{-2}}$ over $100\,\mathrm{m}$ altitude, which is similar to a cloudless case.





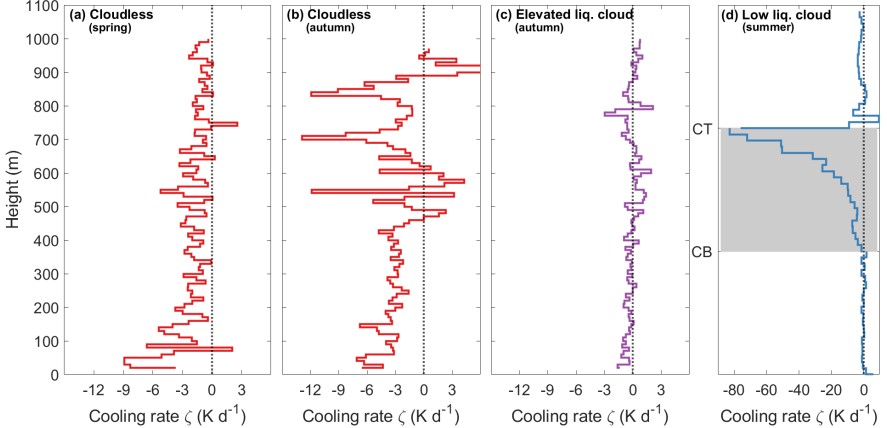

**Figure 5.** Profiles of averaged TIR cooling rates during different atmospheric states: (a) cloudless in spring, (b) cloudless in autumn, (c) under elevated liquid-bearing clouds in autumn, (d) and through a vertically normalized low-level liquid-bearing cloud in summer (note also the different scale on the x-axis for this case). The calculated uncertainty of each data point is $\pm 5\,\mathrm{K\,d^{-1}}$. In (d) cloud boundaries are indicated by grey shading.

The net irradiance beneath low-level (blue lines in Fig. 4) and elevated liquid-bearing clouds (purple lines in Fig. 4) is rather vertically constant, with an average $F_{\mathrm{net}}$ of about -10 W m$^{-2}$. In these states, the downward emission of the cloud is in near radiative equilibrium the upward emission by the surface, resulting in an almost balanced TIR radiative energy budget. This holds especially for clouds with a low base height as frequently observed during MOSAiC. Profiles that exceed the cloud top of low-level clouds exhibit a sharp decrease of $F_{\mathrm{net}}$ at cloud top. This feature describes the vertical transition from the cloudy state

to the cloudless state. At Ny-Ålesund (Fig. 4b, c), only clouds with cloud top higher than 800 m were observed, partially due to the local topography blocking advected clouds. During MOSAiC (Fig. 4a), the cloud thickness and cloud top altitude were more variable, thus the transitions from cloudy to cloudless state were observed at different altitudes. The clouds with different cloud top altitudes observed during MOSAiC also illustrate that the strength of the decrease of $F_{\mathrm{net}}$ at cloud top depends on cloud top altitude. Given the different slopes of $F_{\mathrm{net}}$ profiles in cloudless and cloudy conditions, the spread between both states

increases with altitude. This spread is reflected in the transition at cloud top and enhances cloud top cooling: given a fixed temperature profile and identical cloud microphysical properties, higher cloud tops will cause stronger cloud top cooling.

## 3.3    Radiative cooling rate profiles

For each measured profile of $F_{\mathrm{net}}$, radiative cooling rates were calculated based on Eq. (2). The radiative cooling rates were then averaged for each atmospheric state and season. To assure sufficient statistical significance, in Figure 5 we display only

the results that included at least 8 profiles. In general, radiative cooling rates between -4 K d$^{-1}$ and -1 K d$^{-1}$ are expected for cloudless cases throughout all latitudes (Suomi et al., 1958; Asano et al., 2004; McFarlane et al., 2007; Thorsen et al., 2013; Cesana et al., 2019), while simulations for an Arctic site indicate weak values of about -1 K d$^{-1}$ (Turner et al., 2018). Cloudless





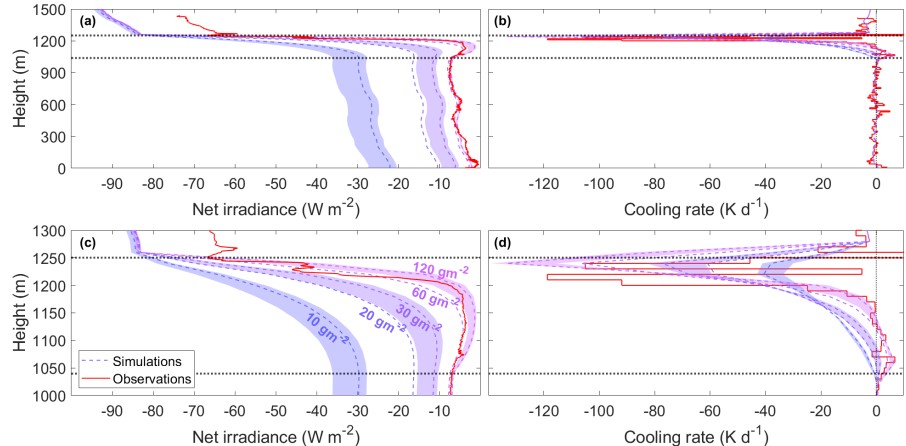

**Figure 6.** Simulated TIR profiles for a single layer low-level cloud with different LWP and a fixed $N_d$ of $70\,\mathrm{m}^{-3}$ (dashed curves). (a) Net irradiance profiles in the lower troposphere, (b) radiative cooling rate profiles in the lower troposphere, (c) net irradiance profiles focusing on the cloud layer, and (d) radiative cooling rate profiles focusing on the cloud layer. For LWPs of $10\,\mathrm{g\,m}^{-2}$, $30\,\mathrm{g\,m}^{-2}$, and $120\,\mathrm{g\,m}^{-2}$, additional profiles obtained by varying $N_d$ between $20–120\,\mathrm{m}^{-3}$ are displayed as shaded areas. Balloon in situ measurements are displayed in red. Cloud boundaries are represented by horizontal dotted lines.

profiles obtained in Ny-Ålesund during spring (Fig. 5a) and autumn (Fig. 5b) show on average a radiative cooling rate between $-3\,\mathrm{K\,d}^{-1}$ and $-1\,\mathrm{K\,d}^{-1}$. Close to the surface, spring measurements have a marked signal up to $-9\,\mathrm{K\,d}^{-1}$. Conversely, up to

about $500\,\mathrm{m}$ height, measurements in autumn indicate slightly larger cooling rates ($-3\,\mathrm{K\,d}^{-1}$) compared to spring ($-2\,\mathrm{K\,d}^{-1}$). Further up, radiative cooling is still present, but the signal becomes more variable due to the surrounding topography.

The radiative cooling rate profile measured beneath elevated liquid-bearing clouds (Fig. 5c) shows weak oscillations typically below $\pm\,1\,\mathrm{K\,d}^{-1}$ centered on the zero line. This case corresponds to the radiative equilibrium where surface and cloud base emit a similar amount of radiation, as previously shown for similar cloud conditions on sea ice (Egerer et al., 2019) and in

Ny-Ålesund (Becker et al., 2020). The average profile obtained through low-level water-bearing clouds in summer shows the radiative cooling rates below, in, and above a vertically normalized cloud (Fig. 5d). Profiling through the cloud layers obviously combines the cloudy and cloudless features, thus weak cooling rates of $-2\,\mathrm{K\,d}^{-1}$ are observed above the cloud and little to no cooling underneath it, while strong cooling rates up to $-80\,\mathrm{K\,d}^{-1}$ are present at cloud top. Similar profiles for Arctic low-level clouds were simulated by Turner et al. (2018) and agree with other in situ observations (Egerer et al., 2019; Becker et al.,

225 2020).

### 3.4 Adjusting measured and simulated radiative profiles

Radiative transfer simulations were performed to quantify the influence of liquid water path (LWP) and droplet number concentration ($N_d$) on the radiative cooling rates of low-level liquid-bearing clouds. An in situ profile obtained on 13 July 2020 is compared to RTMs initialized to represent the observed cloud. During the profile, the LWP retrieved by Cloudnet varied



between 18–49 $\mathrm{g\,m^{-2}}$, with an average of 27 $\mathrm{g\,m^{-2}}$. A droplet number concentration of 70 $\mathrm{cm^{-3}}$ was estimated from the difference of the aerosol particle concentration below and within the cloud, assuming that this is the fraction of activated particles. Cloudnet indicated a cloud between 1040–1260 m. For the simulations, the LWP was varied between 10–120 $\mathrm{g\,m^{-2}}$. The corresponding profiles of liquid water content (LWC) were derived by vertically distributing the liquid water path using an adiabatic LWC assumption. The $N_{\mathrm{d}}$ was varied between 20–120 $\mathrm{cm^{-3}}$. The profile of the droplet effective radius ($R_{\mathrm{eff}}$) of the cloud was then calculated using:

$$R_{\mathrm{eff}}(z) = \left[ \frac{3}{4 \cdot \pi \cdot \rho_{\mathrm{w}} \cdot N_{\mathrm{d}} \cdot k} \cdot LWC(z) \right]^{\frac{1}{3}} \qquad (3)$$

where $\rho_{\mathrm{w}}$ is the density of liquid water and $LWC(z)$ is the liquid water content for the layer at height $z$. The parameter $k$ converts the effective into the volumetric droplet radius and was set to 0.8, which is representative for stratocumulus (Brenguier et al., 2011).

Figure 6 compares the measured with the simulated profiles of $F_{\mathrm{net}}$ and cooling rates for five different LWPs. The simulations show that below the cloud the emission of TIR radiation is quickly saturated with LWP exceeding 30 $\mathrm{g\,m^{-2}}$ (Fig. 6a), consistent with the findings of Shupe and Intrieri (2004), while weak changes of in-cloud $F_{\mathrm{net}}$ occur also at larger values of LWP (Fig. 6c). For thin clouds (LWP below 30 $\mathrm{g\,m^{-2}}$), an increase in the LWP offsets almost homogeneously the $F_{\mathrm{net}}$ in the layer between surface and cloud base, thus without significant variations on the radiative cooling rate. The offset becomes smaller as the LWP increases. Simulated values are about 12 $\mathrm{W\,m^{-2}}$ for LWP increasing from 10 $\mathrm{g\,m^{-2}}$ to 20 $\mathrm{g\,m^{-2}}$, and about 4 $\mathrm{W\,m^{-2}}$ when increasing from 20 $\mathrm{g\,m^{-2}}$ to 30 $\mathrm{g\,m^{-2}}$ (Fig. 6b). In the central layers of the cloud it is still possible to discern a $F_{\mathrm{net}}$ offset due to the increase in the LWP even beyond the 30 $\mathrm{g\,m^{-2}}$ threshold value, while the $F_{\mathrm{net}}$ at cloud top is fixed due to its dependency on the temperature. The overall result is an increased slope for the $F_{\mathrm{net}}$ profile in the cloud top region (Fig. 6c), which causes a marked increase in the cloud top cooling rate (Fig. 6d). For the explored values, $N_{\mathrm{d}}$ plays a role in offsetting the net irradiance in and below the cloud. The effect is stronger below optically thin clouds, with variations up to 7 $\mathrm{W\,m^{-2}}$ at 10 $\mathrm{g\,m^{-2}}$, and becomes weaker with increasing LWPs (up to 4 $\mathrm{W\,m^{-2}}$ at 30 $\mathrm{g\,m^{-2}}$, up to 0.5 $\mathrm{W\,m^{-2}}$ at 120 $\mathrm{g\,m^{-2}}$) (Fig. 6a). The net irradiance offset decreases less markedly inside the cloud (Fig. 6c), and the $F_{\mathrm{net}}$ variability then influences the radiative cooling rates (Fig. 6d).

The maximum radiative cooling rates at cloud top and the integrated radiative cooling of the cloud are calculated in Figure A1. The integrated radiative cooling rate of the cloud layer is calculated by the sum of all the in-cloud radiative cooling rates. Cloud top radiative cooling between -43 $\mathrm{K\,d^{-1}}$ and -33 $\mathrm{K\,d^{-1}}$ is present even with limited amounts of liquid water (Fig. A1a). The magnitude of the radiative cooling increases with the LWP, but for optically thick clouds (LWP larger than 30 $\mathrm{g\,m^{-2}}$) an increase of the available water is not as effective as for thin clouds, and the $N_{\mathrm{d}}$ becomes gradually more important. The integrated cloud radiative cooling is driven almost exclusively by the LWP (Fig. A1b). This occurs since the enhanced cooling provided at cloud top by larger $N_{\mathrm{d}}$ is balanced by reduced cooling in the middle and lower layers of the cloud (Fig. 6d), in agreement with the simulations by Williams and Igel (2021).



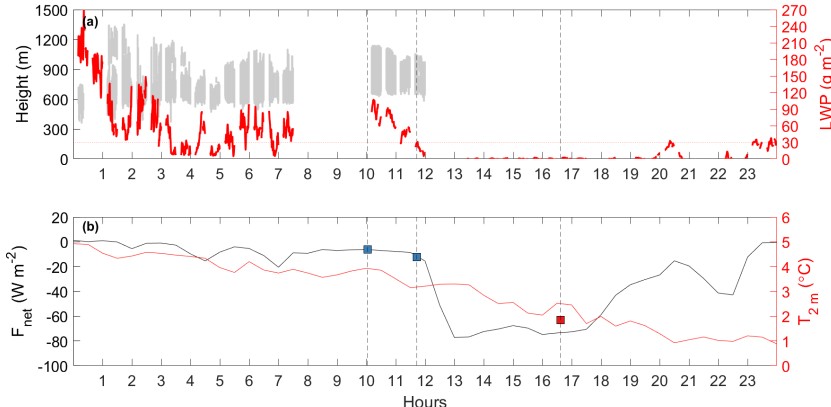

**Figure 7.** Case study on 30 September 2021. (a) Time series of cloud cover (grey areas) and LWP (red line) retrieved by Cloudnet. Between 7:30 and 10 UTC no Cloudnet data are available, periodic gaps are due to the measurement/scan pattern of the microwave radiometer used by Cloudnet to retrieve LWP (see e.g., Chellini et al., 2023). The threshold for optically thick clouds (LWP = 30 g m$^{-2}$) is indicated by a dotted red line. (b) Surface net irradiance (black line) and 2 m air temperature (red line) measured at ground. Near-surface BP measurements of net irradiance at the start of balloon profiles (dashed lines) are indicated by color-coded squares.

## 4 Transition between states

Transitions between atmospheric states may occur due to local processes in the ABL, by the advection of different air masses, and/or by clouds at higher altitudes. Two case studies with consecutive BELUGA observations combined with radiative transfer
simulations were selected to follow the evolution of radiation profiles during such transitions.

### 4.1 Cloudy to cloudless transition

A transition from a cloudy to a cloudless atmosphere was observed on 30 September 2021 in Ny-Ålesund. Figure 7 displays time series of cloud cover and LWP derived from Cloudnet, and surface-based observations of temperature and net irradiance. During the first half of the day, Cloudnet observations showed a low-level liquid-bearing single-layer cloud with cloud base at
around 600 m and cloud top varying between 900–1200 m (Fig. 7a). The LWP was variable but for most of the time exceeded 30 g m$^{-2}$, thus the cloud can be considered optically thick for TIR radiation. Cloudnet data indicated a small fraction of ice particles in the low-level cloud. Radiative transfer simulations (not shown here) illustrated that the ice crystals had no significant impact on the TIR radiation profiles. The low-level cloud started dissipating at around 12 UTC, while the LWP steadily decreased after 10 UTC. Cloudnet also observed a persistent ice cloud with a base height of 7400 m (not shown).
While the low-level cloud was present, the surface time series indicates a constant net irradiance and a temperature decrease on the order of -2.9 K d$^{-1}$ (Fig. 7b). The dissipation of the cloud strongly impacted the surface radiation budget and the resulting temperature trend. The net irradiance became strongly negative due to a reduced downward component in the cloudless atmosphere, resulting in an enhanced temperature decrease at the surface on the order of -5.3 K d$^{-1}$.



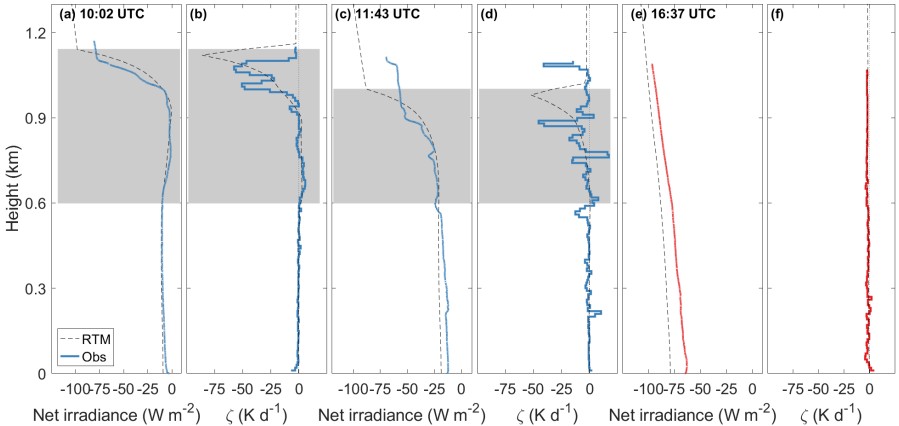

**Figure 8.** Balloon-borne profiles of TIR net irradiance and radiative cooling rates on 30 September 2021 at (a, b) 10:02 UTC, (c, d) 11:43 UTC, and (e, f) 16:37 UTC. Dashed lines represent RTM simulations. Clouds are represented by grey shading.

The profiles of net irradiance and derived radiative cooling rates measured with BELUGA are shown in Figure 8. The
first profile (starting at 10:02 UTC) shows the presence of a cloud layer located between 660–1140 m. A successive profile
observation (initiated at 11:43 UTC) was performed through the dissipating broken cloud layer. Finally, the third profile was
obtained at 16:37 UTC, in a cloudless atmosphere.

The initial in situ measurements of net irradiance and radiative cooling rate (Fig. 8a,b) at 10:02 UTC represent the case
of a low-level liquid-bearing single-layer cloud. When moving upward, the $F_{net}$ decreases significantly at cloud top by about
$70\,\mathrm{W\,m^{-2}}$, which leads to a local maximum radiative cooling rate of $-60\,\mathrm{K\,d^{-1}}$. Low radiative heating rates (up to $5\,\mathrm{K\,d^{-1}}$)
are observed at cloud base as a result of the cloud being colder than the surface. Consequentially, a radiative cooling rate of
$-7\,\mathrm{K\,d^{-1}}$ was calculated for the lowermost layer of the balloon profile. It is important to note that the strong cloud top cooling
is one component of the cloud system. That cooling helps to drive vertical mixing that distributes some of the cooling across
the cloud-driven mixed layer, where the virtual potential temperature is approximately constant (Morrison et al., 2012). Thus,
the temperature change in the cloud is determined by the combined local radiative forcings (cooling at cloud top, warming at
cloud base) and by a collection of additional temperature tendencies including turbulence, phase change, and advective.

The second balloon-borne profile was obtained through a broken cloud layer. Due to the variability of the cloud during
the profile, the net irradiances and thus the derived radiative cooling rates do not show the typical pattern expected for a
homogeneous cloud. In particular, the observed strongest radiative cooling rate is not located at the average cloud top indicated
by Cloudnet, but has a comparable magnitude of $-45\,\mathrm{K\,d^{-1}}$ (Fig. 8d). Despite the radiative warming indicated by the first
profile, the temperature at cloud base (not shown) decreased by $-0.4\,°\mathrm{C}$ in about $1.5\,\mathrm{h}$ ($-6.4\,\mathrm{K\,d^{-1}}$), similarly to the surface
temperature variation, possibly indicating the advection of colder air which cooled the entire ABL.

At 16:37 UTC, the third balloon profile indicates a radiative cooling rate on the order of $-2\,\mathrm{K\,d^{-1}}$ at all heights. The radiative
cooling rate observed throughout the cloudless atmosphere is well represented by the RTM simulation (Fig. 8f). The magnitude



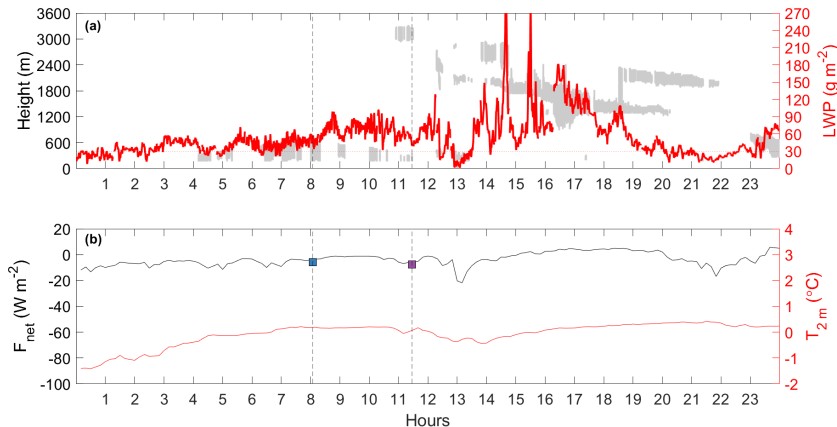

**Figure 9.** Case study on 23 July 2020. (a) Time series of cloud cover (grey areas) and LWP (red line) retrieved by Cloudnet. The threshold for optically thick clouds ($LWP = 30 \, g \, m^{-2}$) is indicated by a dotted red line. (b) Surface net irradiance (black line) and 2 m air temperature (red line) measured on the ice. Near-surface BP measurements of net irradiance at the start of balloon profiles (dashed lines) are indicated by color-coded squares.

of the modeled cooling is lower than the observed trend in near-surface temperature, likely because additional processes like advection of colder air contribute. A negative offset of net irradiance between measurements and simulations is obvious in Figure 8a,c,e and results from the presence of the elevated ice cloud which was not included in the simulation and which effect is analyzed in Sect. 5.2.

### 4.2 Low-level cloud to elevated cloud transition

A transition from a single low-level to a multilayer elevated cloud was observed at the MOSAiC ice camp on 23 July 2020. Figure 9 displays time series of Cloudnet retrieval results, and observations of temperature and net irradiance at 2 m height. A low-level single-layer cloud was observed in the early hours of 23 July 2020 (grey shaded area in Fig. 9a). The cloud was liquid-bearing and optically thick. Cloudnet indicated that, at about 11:00 UTC, the low-level cloud started to be overridden by a liquid-bearing cloud located at about 3000 m. With the advection of the elevated cloud, the low cloud started to dissipate,

and by about 14 UTC the low-level cloud completely disappeared, leaving only the higher thick cloud layer.

Despite the significant variations in the vertical structure of the cloud cover, the surface radiative budget did not significantly change (Fig. 9b). This indicates that the emission of downward irradiance by the cloud base was almost similar for the low-level cloud and the elevated cloud. As a result, the near-surface temperature remained roughly constant throughout the day.

Profiles illustrating the vertical structures of $F_{net}$ and radiative cooling rates during two tethered balloon flights are shown in

Figure 10. The low-level cloud was surveyed by an in situ profile at 08:05 UTC. At 11:28 UTC a second balloon profile was obtained reaching an altitude of about 1000 m, thus profiling through the entire low-level cloud but not the elevated cloud at the time.





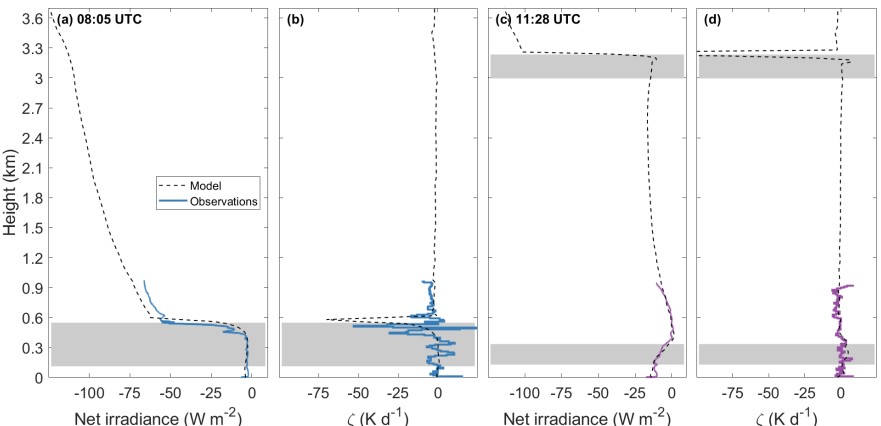

**Figure 10.** Balloon-borne profiles of TIR net irradiance and radiative cooling rates on 23 July 2020 at (a, b) 08:05 UTC and (c, d) 11:28 UTC. Dashed lines represent RTM simulations. Clouds are represented by grey shading.

The first balloon profile clearly shows the radiative profiles of a liquid single-layer cloud, with an in-cloud decrease of net irradiance from -3 W m$^{-2}$ to -55 W m$^{-2}$ (Fig. 10a), resulting in radiative cooling rate of -53 K d$^{-1}$ at cloud top. The value by the RTM is larger due to a 25% stronger net irradiance in the simulated cloud, which uses the cloud boundaries retrieved by Cloudnet. For both observations and simulations, the atmosphere above the cloud has a radiative cooling rate of about -2 K d$^{-1}$.

Radiative transfer simulations were set up to match the observations at 11:28 UTC. The best agreement was found for a LWP of 20 g m$^{-2}$ in the upper cloud and LWP of 20 g m$^{-2}$ in the low-level cloud (Sect. 5.1). The presence of the upper cloud balanced almost completely the upward irradiance from the low-level cloud, where the net irradiance increased from -55 W m$^{-2}$ to about 0 W m$^{-2}$. As consistently shown by both measurements and RTM simulations, this resulted in the complete removal of the strong cooling rate that typically characterizes the cloud top region in single-layer clouds illustrated in Figure 10b. The reduction of cloud top cooling leads to the absence of the turbulent mixing that entrains moisture and aerosol particles from above the cloud (e.g., Morrison et al., 2012; Shupe et al., 2013; Egerer et al., 2019, 2021; Lonardi et al., 2022). Additionally, the positive TIR cooling rates measured at cloud top indicate a weak warming up to 4 K d$^{-1}$, which may cause evaporation. We hypothesize that the combination of enhanced evaporation and the lack of entrainment eventually resulted in the dissipation of the low-level cloud, but the role of advection in this transition is not currently known.

# 5 Impact of elevated clouds

Elevated clouds affect the radiation profiles in the lower troposphere, as shown by the observations in both of the case studies examined here. This effect strongly depends on the optical depth of the elevated cloud. While ice clouds (Sect. 4.1) do not significantly reduce the cloud top cooling of lower clouds, liquid clouds can completely eliminate it (Sect. 4.2). To systematically investigate how the elevated clouds change the radiative cooling rate profiles, radiative transfer simulations are used.





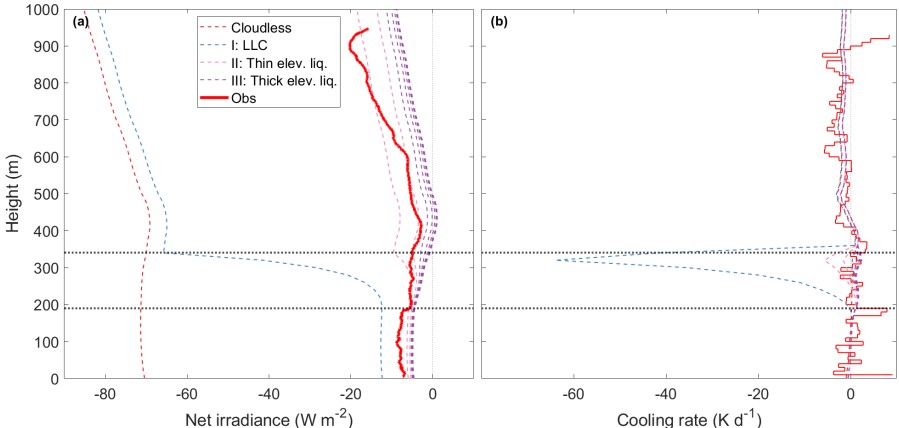

**Figure 11.** Profiles of (a) net irradiance and (b) radiative cooling rates for a liquid-bearing low-level cloud capped by an elevated liquid-bearing cloud. A cloudless simulation is represented in dashed red, a simulation with only the low-level cloud ($\mathrm{LWP} = 20\,\mathrm{g\,m}^{-2}$) is represented in dashed blue, each dashed pink/purple line represents a simulation after increasing the LWP of the upper cloud by $10\,\mathrm{g\,m}^{-2}$, and observed values are shown in solid red. Cloud boundaries are represented by horizontal dotted lines.

## 5.1 Liquid-bearing cloud

Figure 11 displays simulations based on the cloud layers observed on 23 July 2020 at 11:28 UTC. Cloudnet retrievals indicate a total LWP of $38\text{–}72\,\mathrm{g\,m}^{-2}$. This is the sum of the LWP of the upper cloud (altitude between 3000–3200 m) and of the lower cloud (altitude between 190–330 m). We run RTM simulations varying the total liquid water path ($\mathrm{LWP_{tot}}$) between 20–100 $\mathrm{g\,m}^{-2}$ and distributing it differently between the elevated and the low-level cloud layer ($\mathrm{LWP_{elev}}$ and $\mathrm{LWP_{low}}$, respectively). Leaitch et al. (2016) observed $N_\mathrm{d}$ between 30–100 $\mathrm{cm}^{-3}$ in Arctic clouds, and following the good agreement obtained in Sect. 3.4 we set $N_\mathrm{d} = 70\,\mathrm{cm}^{-3}$ as a realistic assumption for the droplet number concentration. The best match with the observed $F_\mathrm{net}$ profile was derived for simulations with $\mathrm{LWP_{elev}} = 20\,\mathrm{g\,m}^{-2}$ and $\mathrm{LWP_{low}} = 20\,\mathrm{g\,m}^{-2}$. This is consistent with the limited Cloudnet retrievals including both the clouds, which indicated a LWP of around $26\,\mathrm{g\,m}^{-2}$ in the elevated cloud and about $16\,\mathrm{g\,m}^{-2}$ in the low-level cloud.

Three modes (I-III) emerge for the radiative cooling rates in the low-level cloud. (I) In the control case the lower cloud contains all the available liquid water and therefore behaves as a single layer cloud (blue line in Fig. 11), with a cloud top radiative cooling rate of -64 $\mathrm{K\,d}^{-1}$. The net irradiance profile above the cloud follows the cloudless profile (solid red line in Fig. 11) with a small offset due to the higher surface emission compared to the cloud top emission with lower temperature. Therefore, a radiative cooling rate of -2 $\mathrm{K\,d}^{-1}$ is derived. (II) When the elevated cloud is optically thin ($\mathrm{LWP_{elev}} < 30\,\mathrm{g\,m}^{-2}$, pink lines in Fig. 11), the net irradiance above the low-level cloud is significantly reduced, resulting in cloud top radiative cooling of -6 $\mathrm{K\,d}^{-1}$ (-2 $\mathrm{K\,d}^{-1}$) when the upper cloud has a liquid water path of $10\,\mathrm{g\,m}^{-2}$ ($20\,\mathrm{g\,m}^{-2}$). (III) In cases where the elevated cloud is optically thick ($\mathrm{LWP_{elev}} \geq 30\,\mathrm{g\,m}^{-2}$, purple lines in Fig. 11), the radiative signature of the low-level cloud



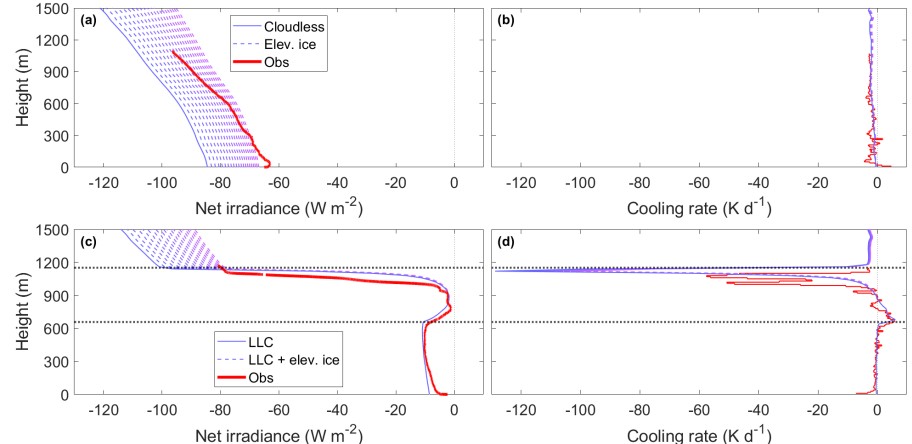

**Figure 12.** Profiles for (a, b) a cloudless ABL and (c, d) a liquid-bearing low-level cloud, both capped by an ice cloud. Net irradiance (left) and radiative cooling rates (right) are displayed. The simulation without the ice cloud is represented by a solid blue line, each dashed line represents a simulation after an increase of the IWP by $1\,\mathrm{g\,m^{-2}}$, and observed values are shown in solid red. Cloud boundaries are represented by horizontal dotted lines.

disappears almost completely. As a result, cloud top cooling of the low cloud is completely removed ($0\,\mathrm{K\,d^{-1}}$), as described by Turner et al. (2018). Thus, the vertical partitioning of LWP in multi-layer cloud situations is very important for the processes impacting the maintenance of all cloud layers.

## 5.2  Ice cloud

Remote sensing retrievals on 30 September 2021 at Ny-Ålesund intermittently showed an ice cloud located between 7400–
9000 m, with a maximum ice water path (IWP) of $20\,\mathrm{g\,m^{-2}}$. We run RTM simulations varying the IWP between $0$–$20\,\mathrm{g\,m^{-2}}$. The ice cloud was investigated for both a cloudy and a cloudless ABL, as displayed in Figure 12. The effective radius of the ice particles was fixed at $30\,\mathrm{\mu m}$ following Cloudnet products.

In general, the variations in $F_{\mathrm{net}}$ induced by the ice cloud are large enough to explain the shifts observed in Sect. 4.1. The ice cloud offsets $F_{\mathrm{net}}$ in a cloudless ABL (dashed lines in Fig. 12a). As the $F_{\mathrm{net}}$ offset is constant with height, the derived radiative
cooling rates do not vary (Fig. 12b). Similarly, the ice layer exerts an effect on a liquid-bearing low-level cloud by diminishing $F_{\mathrm{net}}$ due to an increased downward component emitted by the ice cloud (dashed lines in Fig. 12c). The reduced magnitude of $F_{\mathrm{net}}$ at cloud top adjusts the radiative cooling rates of the low-level cloud (Fig. 12d). For the investigated ice clouds, the maximum effect was a reduction of cloud top cooling from $-129\,\mathrm{K\,d^{-1}}$ to $-100\,\mathrm{K\,d^{-1}}$, and this reduction becomes larger as the IWP of the upper cloud is increased. Overall, it is clear that elevated ice clouds do have the potential to impact the radiative
processes of low-level clouds, but are typically less effective per unit of condensed mass in doing so.



# 6   Conclusions

Vertical profiles of TIR irradiance were measured using broadband radiometers installed within an instrument payload carried by the tethered balloon platform BELUGA during the MOSAiC expedition in summer 2020, and in Ny-Ålesund (Svalbard) in autumn 2021 and spring 2022. The profiles were analyzed to characterize the height-resolved TIR radiative energy budget

and derived cooling rates for typical atmospheric situations. Measurements over the sea ice in summer showed a frequent abundance of low-level liquid-bearing clouds. At Ny-Ålesund mostly cloudless conditions were sampled.

   The measurements show that the typical bimodal distribution of surface TIR net irradiance associated with cloudy/cloudless atmospheric states introduced by Shupe and Intrieri (2004) is present also at higher altitudes. However, the frequency distributions can change significantly with height, as the cloud cover varies with height. Comparing cloudless and cloudy profiles of

$F_{net}$, it becomes obvious that the difference between the cloudy and cloudless states increases with altitude, thus the bimodal distribution becomes more separated. This indicates that the cloud radiative effect is stronger for the atmosphere layers further away from the surface because, while $F_{net}$ is roughly vertically constant below the cloud layer, in cloudless conditions it becomes more negative with height.

   Balloon-borne profiles were divided into four main categories based on the atmospheric state: cloudless, low-level cloud,

and elevated liquid-bearing or ice clouds, each presenting a particular vertical structure of the radiative quantities. Cloudless scenarios exhibit a continuous vertical decrease in $F_{net}$ associated with the decrease in air temperature (Becker et al., 2020; Philipona et al., 2020). The resulting radiative cooling rates are between $-3\,\mathrm{K\,d^{-1}}$ and $-1\,\mathrm{K\,d^{-1}}$ for all observed layers, thus having the capacity to modify the entire temperature profile. The presence of optically thick clouds removes the cooling tendency in the layers between cloud base and the surface. Low-level liquid-bearing clouds are characterized by a strong

cooling rate up to $-80\,\mathrm{K\,d^{-1}}$ at cloud top, which is consistent with previous works (Turner et al., 2018; Becker et al., 2020). Cloud top cooling significantly affects the energy balance of the atmosphere, fueling cloud and turbulence processes that can drive the evolution of the ABL from above (Lonardi et al., 2022).

   Using sensitivity studies based on radiative transfer models combined with the available in situ observations, we showed that measurements were reproduced by simulations by adjusting microphysical input. The liquid water path (LWP) proved

to be of particular importance for the strength of cloud top cooling for optically thin clouds ($\mathrm{LWP} < 30\,\mathrm{g\,m^{-2}}$). For optically thick clouds, the droplet number concentration ($N_d$) acquires relative importance in controlling the magnitude of cloud top cooling, but the integrated cloud cooling does not significantly change with varying $N_d$ (Williams and Igel, 2021). An elevated cloud impacts the lower troposphere radiation profiles similarly to a low-level cloud, with the important distinction that its cloud top cooling is located above the ABL. The presence of an elevated cloud has an impact also on low-level clouds (Turner

et al., 2018). Even when the amount of liquid water in the elevated cloud is limited, the cloud top radiative cooling in the low-level cloud is strongly reduced. The radiative cooling in this lower cloud then becomes negligible when the elevated cloud is optically thick. Even elevated ice clouds can modulate the radiation profiles by reducing the radiative gradient within lower-level clouds, but they are less effective at doing so per unit of condensed mass. In cloudless boundary layers, the reduction of $F_{net}$ due to an ice cloud aloft is important because car vary the surface radiative balance, but does not affect the strength of the



radiative cooling rates over the vertical profile. This reduces the radiative gradient within the low-level cloud, thus dampening the radiative cooling rate at cloud top.

To study what a transition from one atmospheric state to another may cause, two scenarios were analyzed by a sequence of balloon profiles. In both cases, the vertical profiles of the radiative energy budget and the radiative cooling rates changed significantly and rapidly. In the first case (30 September 2021), a low-level cloud dissipated during the advection of a colder

air mass. The removal of the cloud caused a sudden increase in the magnitude of $F_{\text{net}}$ at the surface. This promoted a localized strong cooling which resulted in a surface-based temperature inversion, and a generalized cooling at all heights. In the second case (23 July 2020), an advected elevated cloud led to the dissipation of a low-level cloud by weakening its cloud top radiative cooling. The lack of cloud top cooling is hypothesized to result in less vertical motion, less entrainment, and finally dissipation of the cloud layer. In this case, the total surface radiative energy budget at the surface did not change significantly although the

cloud situation changed. The downward emission of the elevated cloud had the same magnitude as the surface warming of the low-level cloud.

Although the number of observed profiles is limited, the data presented here showcase the radiative cooling rate profiles associated with different atmospheric states, and how transitions between these states take place. However, to fully represent the complexity of such transition scenarios and compare them to numerical models, a larger set of consecutive profile observations

with less temporal separation is recommended.

*Data availability.* The balloon-borne radiation measurements are currently available upon request. The 2 m time series of net irradiance for MOSAiC was obtained by Cox et al. (2023). The 33 m time series of net irradiance for Ny-Ålesund is available at the Italian Arctic Data Center (https://metadata.iadc.cnr.it). Cloudnet data were obtained by Griesche et al. (2023) for MOSAiC and by https://cloudnet.fmi.fi/ for Ny-Ålesund. Radiosonde data were obtained by Maturilli et al. (2022) for MOSAiC and by Maturilli (2020) for Ny-Ålesund.

**Appendix A: Radiative cooling rates for cloud top and the integrated cloud**

*Author contributions.* ML performed the balloon-borne radiation measurements, analyzed the data, and drafted the manuscript. MDS and MM were responsible for the surface radiation measurements at MOSAiC and Ny-Ålesund, respectively, and contributed to the data analysis. EA, AE, CP, HS and MW contributed to the acquisition of balloon-borne measurements and to the scientific discussion.

*Competing interests.* The authors declare that they have no conflict of interest.



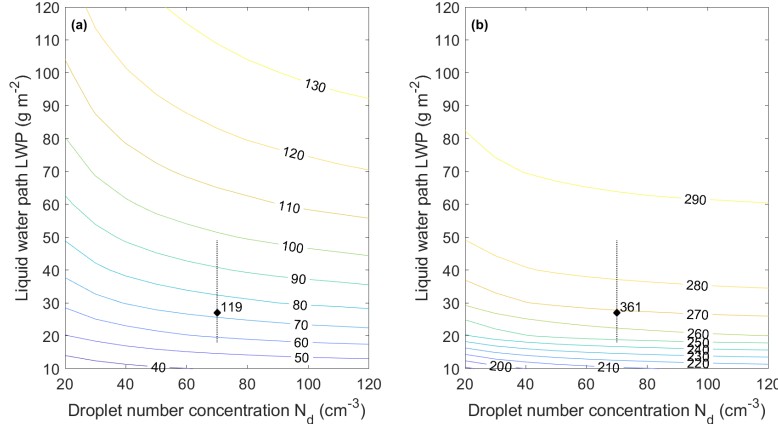

**Figure A1.** Contour plots of (a) cloud top radiative cooling [K d$^{-1}$] and (b) integrated cloud radiative cooling [K d$^{-1}$] with varying liquid water path and droplet number concentration. The black dots indicate the observed radiative cooling rates, and the dotted line indicates LWP variability during the profile.

*Acknowledgements.*  We gratefully acknowledge the funding by the Deutsche Forschungsgemeinschaft (DFG, German Research Foundation) – project number 268020496 – TRR 172, within the Transregional Collaborative Research Center "ArctiC Amplification: Climate Relevant Atmospheric and SurfaCe Processes, and Feedback Mechanisms (AC)[3]" in sub-project A02. This work was carried out and data used in this manuscript were produced as part of the international Multidisciplinary drifting Observatory for the Study of Arctic Climate (MO-SAiC) with the tag MOSAiC20192020. We thank all persons involved in the expedition of the Research Vessel *Polarstern* during MOSAiC

(AWI_PS122_00) as listed in Nixdorf et al. (2021). MDS was supported by a Mercator Fellowship with (AC)[3], by the US National Science Foundation (OPP-1724551), Department of Energy (DE-SC0021341), and NOAA Cooperative Agreement (NA22OAR4320151). Radiation measurements for MOSAiC were obtained from the University of Colorado / NOAA flux team. Radiation measurements for Ny-Ålesund were obtained from the Italian National Research Council. Cloudnet data for MOSAiC were obtained from the Cloudnet team at TROPOS. We acknowledge the assistance from the staff of the Institute for Geophysics and Meteorology of the University of Cologne in the interpre-

tation of the Ny-Ålesund Cloudnet data. Radiosonde data were obtained through a partnership between the leading Alfred Wegener Institute (AWI), the Atmospheric Radiation Measurement (ARM) User Facility, a US Department of Energy facility managed by the Biological and Environmental Research Program, and the German Weather Service (DWD). The authors are particularly thankful to the teams operating BELUGA and to the logistics staff present at MOSAiC and in Ny-Ålesund.





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
