# Peer review of "Tethered balloon-borne observations of thermal-infrared irradiance and cooling rate profiles in the Arctic atmospheric boundary layer"

_EGUsphere, 2023_

## Author Response (AR1)

The manuscript was revised following the comments by the two reviewers. In particular, most parts of Section 4 (Transition between states) were removed. Based on the data used for the case study formerly presented in Section 4.1, a new Section (Profile of cloud radiative effect) was produced. The abstract and conclusions were modified accordingly.

In the following document, we address the comments by the reviewers.

**1   Reviewer 1**

The (appropriately titled) paper, "Tethered balloon-borne observations of thermal-infrared irradiance and cooling rate profiles in the Arctic atmospheric boundary layer" by M. Lonardi et al. provides an interesting analysis of a valuable new dataset, tethered balloon profiles of radiative fluxes and associated parameters obtained during the MOSAiC expedition. The combination of analysis of the tethered balloon profiles with simulations provides a valuable description of the radiative characteristics of the arctic environment. I appreciated the modeling sensitivity tests, which provided some physical understanding of the clouds beyond their radiative properties.

For the most part, I thought that the flow of the paper was logical and appropriately cited previous work. It appears that sufficient information is also provided that would allow someone to attempt to reproduce these calculations. I would have appreciated more of a summary of some of the measurements. I know that details are provided in Lonardi et al. (2022), but the CloudNet retrievals play a central part in the discussion so it would be useful to have at hand a list of the measurements used for those retrievals. Similarly, while the focus of this paper is on radiative profiles, there are places in the paper where information about other measurements would have been useful (e.g. surface temperatures or temperature profiles).

We agree that crucial information on the instrumentation was missing. The following text is now included in the revised manuscript. "The near-surface radiation data of MOSAiC was obtained by a pair (upward and downward looking) of pyrgeometers (uncertainties of $2.6\,\mathrm{W\,m^{-2}}$ and $1\,\mathrm{W\,m^{-2}}$, respectively) located at $2\,\mathrm{m}$ height as part of an Atmospheric Surface Flux Station (ASFS) (Cox et al., 2023). In Ny-Ålesund, the Climate Change Tower (CCT) was equipped with a net radiometer (accuracy $10\,\%$) installed at $33\,\mathrm{m}$ height above ground (Mazzola et al., 2016). The Cloudnet algorithm was applied to calculate profiles of cloud properties by combining an ensemble of retrievals from surface-based remote sensing instruments (Illingworth et al., 2007). For MOSAiC, the cloud characterization with Cloudnet was based on a microwave radiometer, a Ka-band cloud radar, and a microwavelength Raman lidar (Griesche et al., 2023). The LWP uncertainty is $20\,\mathrm{g\,m^{-2}}$, while for the IWP is $40\,\%$. The LWC has an uncertainty of $15\,\%$–$20\,\%$, and for the IWC is between -$30\,\%$ and $40\,\%$ (Lonardi et al., 2022). In Ny-Ålesund, the ground-based remote sensing featured a microwave radiometer, an FMCW-94-SP Doppler cloud radar, and a ceilometer (Nomokonova et al., 2019). Vaisala RS41 radiosondes were used at both sites (Maturilli, 2020; Maturilli et al., 2022)."

I generally found the discussion of the net radiative fluxes and cooling rates to be clear. However, there were locations (see specific examples in the line-by-line comments below) where I feel that statements are made about the implications of these net fluxes without regard for other components of the heating budget. I particularly noticed this in section four, which focuses on the transition between atmospheric states. There seemed to be an argument made that the radiative cooling rates were responsible, or partially responsible, for the transition between states. But there are very likely other processes at play (e.g. advection, solar heating, turbulent heat fluxes). From my perspective, I don't think this section adds significantly to the overall analysis and could be removed – unless there is an intent to make a more direct causal relationship between the cooling profiles and the changes in atmospheric state. If that is case, I believe that a stronger case needs to be made.

With the exception of the caveat noted for section four, I found the overall analysis to be useful and the conclusions to be an appropriate summary of the results.

Specific comments tied to line numbers in the document are provided below.

**Comments**

7-8: The magnitude of radiative cooling is given for cloudless and low-level liquid-bearing clouds – but the numbers have different meanings. For the cloudless case, I believe the value is a layer or column average, whereas the value for the low-cloud case is the peak at cloud top. That should be explained or corrected for consistency.

45  We clarified this by changing the text into: "Cloudless cases display an average radiative cooling rate of about -2 K $\mathrm{day}^{-1}$ throughout the atmospheric boundary layer. Instead, low-level liquid-bearing clouds are characterized by a radiative cooling up to -80 K $\mathrm{day}^{-1}$ within a shallow layer at cloud top, while no temperature tendencies are identified underneath the cloud layer."

  16-17: I am uncomfortable with the statement: "The enhanced warming in this region is a result of different feedback mechanisms known as Arctic amplification". The references listed given provide good discussions of this topic. I think it would
50 be more correct to say that the enhanced warming in the arctic relative to the global average is known as arctic amplification – and the cause for this phenomena, though not yet fully understood, includes several feedback mechanisms and other processes.

  We agree and changed the section following the suggestion: "The Arctic climate system is currently undergoing dramatic changes that are mainly driven by the global warming. However, in the Arctic the warming is significantly enhanced relative to the global average. This effect is known as "Arctic amplification". The causes cause for this phenomenon, though not yet fully
55 understood, include several feedback mechanisms and other processes (Serreze and Barry, 2011; Wendisch et al., 2023)."

  18: What is meant by "In this framework"? What framework is being referred to? I think it is fair to say that representation of clouds in most environments includes uncertainties.

  We agree with the reviewer that the representation of clouds is generally uncertain. However, we want to stress that, with the cloud being part of several feedback mechanisms in the AA, their uncertainty becomes particularly relevant. The text
60 was changed to: "Among others, the role of clouds in Arctic amplification and their realistic representation in models is still uncertain, mainly because of the complexity of cloud processes (Curry, 1986; Curry et al., 1996; Morrison et al., 2012). Reducing the uncertainty is of primary interest, as clouds play a crucial role in several feedback mechanisms (Wendisch et al., 2019). In particular, clouds exert a strong control on the Arctic surface radiative energy budget (Intrieri et al., 2002; Becker et al., 2023)."

65  14: I think the text would be more correct if modified to read: "demonstrating the (greater) radiative significance of the liquid clouds (relative to ice clouds)". Or something that conveys the fact that liquid and ice cases are being compared here.

  We agree and changed the text following the suggestion of the reviewer. "Additional radiative transfer simulations are used to demonstrate the enhanced radiative importance of the liquid relative to ice clouds."

  76: The text indicates the MOSAiC camp was "placed" on an ice flow with melting snow. It was located in such conditions
70 in the summer – but it was originally placed in moderately thick ice in the late fall and allowed to freeze into the floe over the winter.

  The reviewer makes a correct point. The text was changed to clarify that the conditions described in the text were the ones relative to the summer, as the balloon was not deployed earlier in the season. "When BELUGA was deployed in summer 2020, the MOSAiC camp consisted of an ice floe covered with melting snow, surrounded by an increasing fraction of melt ponds and
75 open water."

  95: The solar component is noted to be of secondary importance and this is likely true; however, in some later conclusions, I am wondering how true this is – especially during summer – and whether some of the residual effects could be due in part to the neglected solar component. Has this been explored?

  Indeed solar radiation plays a non-negligible role in the radiative balance of Arctic, as we briefly introduce in lines 96–97.
80 However, at cloud top, the radiative cooling by emission of thermal infrared radiation is dominating. Here we refer to the simulations by Turner et al. (2018). Therefore, we did not include own simulations for solar radiation in the analysis and focused on the thermal IR effect of the clouds. As we removed the discussion on causes for the transition between cloud states, it is now not needed anymore to compare the relevance of all contribution processes such as solar warming and evaporation.

  100: Is the longwave uncertainty really 7 W/m$^2$ regardless of the actual irradiance? I would have thought it would be a
85 percentage of the absolute value.

  Yes, the value of 7 W/m$^2$ was estimated as the maximum uncertainty to be expected for the highest irradiance values. For lower irradiances, the uncertainty of course is lower.

  115: I believe plan-parallel should be plane-parallel

  Corrected the typo.

90  132-134: The text states that the summer atmosphere over ice is "typically covered by fog and/or liquid-bearing low clouds" but the text then goes on to say that the summer of 2020 was unusually warm resulting a "lower share of ice-bearing clouds in favor of purely liquid-water clouds". This seems to suggest that the absence of ice is atypical – in contradiction to the earlier description.

We agree that the text was poorly formulated. The intent of the second part was to highlight the unusual prevalence of pure liquid clouds in contrast to the typically present mixed-phase clouds. The text was changed as follows: "The Arctic summer atmosphere over the pack ice is typically covered by fog and/or low-level clouds, mostly in the form of mixed-phase clouds (Curry et al., 1996; Shupe, 2011; Tjernström et al., 2012; Brooks et al., 2017). Rinke et al. (2021) showed that summer 2020 was unusually warm at the MOSAiC site, resulting in a lower share of mixed-phase clouds in favor of purely liquid-water clouds."

138: The text states that autumn clouds are typically centered at about 1km height. I'm not clear what this means. Does this mean that cloud bases are typically around 1km? Or that the mid-point between cloud base and cloud top is typically around 1 km? Does this only refer to single-layer boundary layer clouds?

The 1 km altitude does not refer to cloud base. It results from the vertically resolved frequency distribution of hydrometeors published by Nomokonova et al. (2019). This analysis indicates that clouds in NyA would typically be located on the upper edge of BELUGA profiles, or possibly outside its reach. It is valid for all types of clouds and hydrometeors. We also included here a revised sentence on the effect of the topography. To clarify our statement, we changed it to: "A statistical analysis conducted by Nomokonova et al. (2019) showed that Ny-Ålesund is generally characterized by the presence of clouds throughout the entire year (81 %). In autumn the vertical frequency distribution of hydrometeors is typically centered around a maximum at about 1 km height, while clouds at lower altitudes are less frequent as the topography prevents the advection of low maritime clouds into the valley (Maturilli and Kayser, 2017). This height is close to the ceiling height for tethered balloon profiles, thus it was expected that BELUGA measurements often stop below cloud top. Due to the not sampled cloud top, the corresponding radiative cooling profiles do not show the cloud top cooling and thus are classified as elevated cloud cases."

139-140: I don't think it is appropriate to refer to 13 cases as "a vast presence"

We agree that, given the number presented, the formulation was exaggerated. We rephrased indicating: "BELUGA data in autumn 2021 consistently showed a larger number of elevated clouds (13 cases) compared to low-level clouds (four cases)."

140: The text states that low-level clouds were "inhibited by local topography" but no explanation is offered by how low clouds are inhibited. I have often seen clouds form in valleys – so it is not clear what is being referred to here. I believe further explanation is needed.

We agree with the reviewer, our phrasing was oversimplified. Maturilli and Kayser (2017) showed in their Figure 1 that, excluding the first km above NyA, the circulation is directed along a westerly flow. As the clouds formed over the ocean are advected along this direction, the local orography prevents the advection of low-level clouds but allows for the passage of what is higher than this barrier. Since the ceiling for balloon flights was typically in the same range of altitudes, most of the advected clouds encountered at the site would then be classified as elevated clouds.

141: It may be a question of semantics, but I think it would be more correct to say that cloudless cases were the dominant state sampled in spring rather than "observed" There appear to be plenty of cloudy cases in the spring, but they didn't occur at a time that BELUGA was flying (e.g. the middle part of April)

The formulation suggested by the reviewer clarifies our statement, which indeed was referring to the balloon flights. "Observed" changed to "sampled".

171: Could the difference between Ny-Alesund and summer (MOSAiC) be due in part to differences in the surface (land vs. ocean/sea ice)?

In general yes (e.g, Figure 5e,f in Wendisch et al. (2022)) but there are reasons why it is not the case for this study. Both the MOSAiC ice floe and the ground site at NyA were covered in snow (besides the first three weeks of the autumn campaign, when only 2 cloudless flights were obtained). Indeed the snow was in different conditions since at MOSAiC it was melting, in autumn at NyA it was accumulating, and in spring it was roughly stable. Therefore, as the upward radiation emitted by two surfaces at different temperatures would be different, there is an effect caused by the surface, but in our opinion, this falls again into the "temperature difference among seasons" explanation. However, it is good to explicitly address this point. We included the following sentence: "The temperature variations affected both the downward irradiance emitted by the atmosphere and the upward irradiance emitted by the surface."

175-176: I would expect that variations in in the profile top Fnet would be due to the superposition of the cloud cover variation and variation in the vertical thermodynamic profile.

Indeed the discrepancies between the near-surface and the top-of-profile balloon observations are potentially caused by the combined effects of vertical variation of the cloud cover and temporal variations of the cloud cover. However, the surface

observations at these time steps (dotted lines in Figure 3a,d) do not exhibit significant variations, implying that the overlaying atmosphere has not changed. The variation reported by the balloon profiles (colored lines), instead, indicates a variation in the cloud cover at the top-of-profile.

176-177: The text states that MOSAiC "mostly featured only low-level cloud". To me, the top-of-profile distribution of Fnet looks more flat across the range of Fnet. There is a peak around -60 W/m$^2$ but there are a significant number at or above -30 W/m$^2$ as well.

The statement indicated by the reviewer was meant to highlight the difference between the balloon observations near the surface (Fig. 3a, red bars) and the ones at the top-of-profile (Fig. 3d, blue bars). In comparison to the NyA profiles, where there is typically no shift through the profile (which indicates either cloudless cases or elevated liquid-bearing clouds), the significant transition from cloudy-like to cloudless-like irradiances at MOSAiC is a proxy for the crossing of cloud top. However, the reviewer has a point, as from Table 1 it can be noted a relative abundance of elevated liquid-bearing clouds as well. The text was changed from "mostly featured only low-level clouds" to "often featured single layers of low-level liquid-bearing clouds".

189-190: I am wondering whether gradients in emissivity due to gradients in water vapor also play a role in the Fnet profile.

In a cloudless atmosphere, a layer with increased humidity has a radiative effect, as we now state in the new Sect. 5. In a cloudy case, the emission by the cloud dominates and covers this effect.

198: I don't think this is a correct usage of the term "radiative equilibrium". I believe that radiative equilibrium refers to the state where the radiative flux absorbed by a volume is equal to the radiative flux emitted by the volume. In the case here – a downward flux is stated as being equal to the upward flux. I don't think that guarantees radiative equilibrium.

We agree that the terminology was misused. We changed the text to: "In these states, the downward emission of the cloud has a similar magnitude to the upward emission by the surface, resulting in an almost balanced TIR radiative energy budget."

205-206: Would the assertion about cloud top cooling and its variation with the height of cloud top still be true as the cloud top was varied across a temperature inversion?

If cloud top was lifted into a temperature inversion, $F^{\uparrow}$ at cloud top would increase due to the increased temperature, thus increasing the negative magnitude of the $F_{net}$ at cloud top height and then increasing the cloud top radiative cooling rate.

216: The text states that the "(cooling) signal becomes more variable due to the surrounding topography." What is the basis for concluding that topography is responsible for cooling rate variability? It seems that there could be other explanations such as variability in the thermodynamic profile.

The reviewer makes a correct point. The reason is in fact a variation of the thermodynamic profile at heights compatible with the crossing of the topography. The air above such altitude level would typically be an advected air mass, with different properties from the air contained in the fjord. We stated it explicitly: "Further up, radiative cooling is still present, but the signal becomes more variable due to the thermodynamic profiles of the air masses advected above the surrounding topography." We also realized that the text at the beginning of the paragraph was misleading, as the number of profiles included in the figure decreased with height. We reformulated as follows: "To assure sufficient statistical significance, the averages shown in Figure 5 were calculated only when at least 6 profile observations were available in the corresponding altitude." Figure 5 was changed accordingly.

234: Are the cloud boundaries assumed to be fixed at those observed by CloudNet?

The cloud boundaries follow the Cloudnet retrievals (1040–1260 m) while the lines previously shown in the Figure represented the in-situ observations (1040–1250 m). To be consistent with the RTM, the cloud boundaries in the Figure were modified to match the Cloudnet retrievals.

244: I am not following the meaning of the statement: "an increase in the LWP offsets almost homogeneously the Fnet in the layer between surface and cloud base"

Our initial statement was not sufficiently clear, and was changed to: "For thin clouds (LWP below $30\,\mathrm{g\,m^{-2}}$), an increase in the LWP increases $F^{\downarrow}$ at cloud base. This increase offsets the $F_{\mathrm{net}}$ by a constant amount at all the heights in the layer between surface and cloud base. Due to the constant offset, the slope of $F_{\mathrm{net}}$ remains unchanged and no significant variations of the radiative cooling rate occur."

249: The statement is made that "Nd plays a role in offsetting the net irradiance in and below cloud". But doesn't that depend on how Nd is varying relative to the LWP?

The reviewer raises a correct point, however, we meant variations of droplet number concentration at fixed LWP. In the simulations, we varied the LWP and the droplet number concentration independently. We presented the variation due to LWP at a fixed Nd and then the variation induced by Nd at a fixed LWP. In particular, we use the change of the droplet number concentration to investigate the potential indirect effect of a variation in the aerosol particles acting as cloud condensation nuclei. We edited the text to: "For the explored values, a variation in $N_d$ at a fixed LWP (shaded areas in Fig. 6) can change the net irradiance in and below the cloud."

254-261: I have a few comments regarding this section on maximum and integrated cooling rates. To begin with, I am not clear why the figure is an appendix when the discussion is in the body of the article. I would suggest either fully integrating this discussion into section 3.4 and moving the figure there – or moving the discussion to the appendix with more explanation on how it amplifies the article. I am also unclear on the concept of integrated cooling rate. This is not a quantity that I have encountered before. I looked at the cited reference (Williams and Igel, 2021) and it is still not clear there how the quantity adds to the understanding of the cloud – but I would also note that Williams and Igel use different units for integrated cooling vs. the local (cloud top) cooling. I suggest that if you are going to present the integrated cooling, you provide some discussion of the importance of the quantity and verify that the units are appropriate.

We agree with the reviewer, the Figure was integrated into the body of the article after an improvement for readability. Regarding the second point, we used the term "integrated cooling rate" as a quantification of the cooling within the entire cloud layer assuming the radiative imbalance is cooling the entire layer. The purpose was to show that an increase in the cooling maximum (at cloud top) does not imply necessarily an enhanced cooling for the full cloud layer. This analysis has two motivations. First, the magnitude of the cloud top cooling rate depends on the layer thickness chosen for the calculation. Distributing the cooling to the entire cloud makes the numbers more comparable. Second, in reality, when time and dynamics are included in the interpretation, the cloud top cooling will be distributed within the cloud. Induced vertical motion leads to turbulent mixing. Thus, the cloud top cooling will cool the entire cloud. This distinction is important, as the latter term is one of the quantities regulating the temperature tendency of the air mass, while the former is driving the cloud top entrainment. We modified the text as follows: "While the maximum radiative cooling plays an instantaneous crucial role in driving the entrainment at cloud top, the radiative cooling of the entire cloud layer ("integrated cloud radiative cooling") characterizes the long term temperature tendency of the cloudy air mass. The integrated cloud radiative cooling of the cloud layer is calculated by the sum of all the in-cloud radiative cooling rates divided by the cloud thickness. However, it has to be noted that other terms would significantly impact this temperature tendency (solar radiative warming, phase change, advection, turbulence)."

The integrated cloud radiative cooling can be also calculated in $\mathrm{W\,m^{-2}}$ following our eq. 2 and assuming $F_{\mathrm{net}}(z_{\mathrm{bot}}) = 0\,\mathrm{W\,m^{-2}}$. The result should then be multiplied by 86400 to convert from seconds to day. More specifically, converting the value of about $85\,\mathrm{W\,m^{-2}}$ obtained by Williams and Igel (2021) in their Figure 3a yields an integrated cloud radiative cooling of $22.6\,\mathrm{K\,d^{-1}}$. Williams and Igel (2021) chose to express their result in $\mathrm{W\,m^{-2}}$, which has the advantage of making it independent of the cloud thickness ($250\,\mathrm{m}$ in their case). As we calculated this for a specific cloud, we believe it is appropriate to maintain the result in the form of $\mathrm{K\,d^{-1}}$, in particular because of the non-zero TIR net irradiance observed at cloud base.

277-78: The statement is made that "The net irradiance became strongly negative due to a reduced downward component in the cloudless atmosphere, resulting in an enhanced temperature decrease at the surface on the order of -5.3 K d$^{-1}$". I'm not sure I believe this cause and effect. There may be other factors impacting the temperature trend including shortwave fluxes, turbulent heat fluxes, or advection. There are statements made at other parts of the manuscript (e.g. line 297 or 301) noting the possible role in some of these heating components – so I am surprised by this cause and effect statement here.

We did not address this point as we removed the cause-effect statement from this section.

287: Is the quoted radiative cooling rate of -7 K d$^{-1}$ an average for the below-cloud layer? What is the "lowermost layer"?

The text was not clear and referred to the lowermost observed value, in the near-surface layer, to highlight the interplay between surface and cloud base. The text was changed to: "Consequentially, a radiative cooling rate of -7 K d$^{-1}$ was calculated in the near-surface layer (lowermost $10\,\mathrm{m}$-layer of the balloon profile)."

309: Another cause-effect question – it seems that it is being asserted that the elevated cloud is causing the low cloud to dissipate – is that true? If so, what is the driver? The additional heating from the upper cloud? There must be other factors at play since multi-layer clouds are not unusual.

We did not address this point as the section was removed.

330: What is the basis for this hypothesis? This is another cause-effect situation. As stated – the impact of advection is not known. Is there any indication of what the advective tendencies might be?

338: In Figure 11, there seem to be more lines than I can find definitions for in the text.

The text has been edited to clarify this point: "We run RTM simulations varying the total liquid water path ($LWP_{tot}$) between 20–100 $\mathrm{g\,m^{-2}}$ (in steps of 10 $\mathrm{g\,m^{-2}}$) and distributing it differently between the elevated and the low-level cloud layer ($LWP_{elev}$ and $LWP_{low}$, respectively)."

347: The text mentions "three modes". I would think of modes as three observed states. Two of the modes would seem to be better described as limiting states.

We agree that the terminology was misused. We meant "states" and the text was changed accordingly.

361: It appears that the ice amount in the upper level cloud has no impact on the net irradiance within the cloud. Is that true?

Yes, it is correct that the variation of the IWC/IWP in the elevated ice cloud does not significantly change the TIR net irradiance profile within the low-level cloud layer. This is due to the quick absorption and emission when LWC is sufficiently high. Within the low-level cloud, the irradiance profile is determined by the local thermodynamic and microphysics. However, the presence of the elevated ice cloud changes the net irradiance above the cloud top of the low-level cloud. This changes the gradient of net irradiance, thus it has a dampening effect on the radiative cooling rate of the uppermost layers of the cloud. We made some slight changes to the caption of the Figure: "Profiles for (a, b) a cloudless ABL and (c, d) a low-level liquid-bearing cloud, both capped by an elevated ice cloud (not shown). Net irradiance (left) and radiative cooling rates (right) are displayed. A simulation without the ice cloud is represented by a solid blue line, each dashed line represents a simulation after an increase of the IWP by 1 $\mathrm{g\,m^{-2}}$ in the elevated cloud, and observed values are shown in solid red. The boundaries of the low-level cloud are represented by horizontal dotted lines."

388: The statement is made that the radiative cooling ratees "(have) the capacity to modify the entire temperature profile". I am not clear on the point of this statement. It is typical for there to be non-zero heating/cooling rates through the atmospheric column. This can result in modifying the temperature profile – but it can also have other effects (as is noted in places in the article).

The reviewer is right, our statement aimed at indicating that the radiative temperature tendencies can potentially provide energy at all heights. The text was changed to: "For all cloudless cases, the radiative cooling rates ranged between -3 $\mathrm{K\,d^{-1}}$ and -1 $\mathrm{K\,d^{-1}}$. This loss of radiative energy can potentially modify the entire temperature profile and thus initiate other atmospheric processes."

404: I believe "car" should read "it can" – but I'm not certain.

The typo was corrected.

418: There appears to be another cause/effect statement being made here about the transition between states. It appears (but I am not certain) that an argument is being made that the cooling profiles are responsible (or partly responsible) for the transition between states – but I don't think that is necessarily the case. There could be (and likely are) larger scale processes at work.

This sentence was removed after editing Section 4 (now Section 5).

**2 Reviewer 2**

The authors utilize upward and downward-looking irradiance measurements that were obtained from instruments tethered to balloons to infer radiative heating and cooling rate profiles. The measurements cover multiple Arctic locations and different seasons. They find characteristic differences for clear and cloudy scenes, largely in line with radiative transfer calculations, and highlight the role of multi-layer clouds to suppress cloud-top cooling in the lower layers. The authors explore the role of cloud micro- and macro-physical properties for this suppression.

The paper is well written, and the figures complement the text adequately. I suggest one major and several minor modifications before publication.

**Major concerns**

The paper lacks a discussion section. There are a few points the authors touch on and additional ones that come to mind. I suggest writing a discussion section preceding the summary that includes, for example, the below points:

I like the brief discussion (ll. 326-331) on the dynamical impacts of cloud-top cooling (or the lack thereof). Perhaps the authors could expand on it. Are there other constituents the authors haven't explore in this paper? I could image lofted aerosol as well as lofted moist air could impact boundary layer heating and cooling rates. Based on the profiles used here, can the authors claim full understanding of Arctic radiative transfer or are there remaining deficiencies in RTM simulations?

The main objective of this manuscript is to highlight the type of radiation profiles in the Arctic lower troposphere in function of clouds and their microphysics. Of course, other components can change the heating/cooling rates. However, compared to clouds, the impact of humidity and aerosols is small. We added a statement on the effect of humidity, while we did not discuss aerosol particles as their (direct) effect is expected to be small on TIR radiation (Protat et al., 2014). In order to separate the cloud radiative effect, we added an analysis making use of the state transitions of the original manuscript. Having measurements in cloudy and cloudless conditions, the cloud radiative effect can be calculated and to some part extracted from aerosol and humidity effects. The remaining atmospheric effect was estimated by simulations using two different atmosphere profiles. We removed the discussion regarding the cause-effect during the transition in the radiation profiles, as the section was removed, and we now offer an interpretation of these variations as cloud radiative effect. With the new structure of the analysis, we think adding a separate discussion section is not needed. All relevant discussions of the results are implemented in the corresponding sections and summarized in the conclusion section.

**Minor concerns**

ll. 38-40 This sentence needs additional information. Is free-tropospheric cooling or warming meant here?

We don't understand completely the point raised by the reviewer. The radiative features we indicate are expressed for the lower troposphere, and we wanted to indicate the relevance of radiative fluxes in controlling the evolution of the thermodynamic profiles. In particular, the cloud top TIR radiative cooling has a role in the formation and maintenance of the temperature inversion (Zhang et al., 2020), thus it strengthens the decoupling between the boundary layer and the free troposphere above.

Section 2: Please explain how aerosol was included in radiative transfer calculations and which aerosol properties were assumed.

As we focused on the TIR component, which is only slightly impacted by the direct effect of aerosols (e.g., Protat et al., 2014), we used the standard aerosol properties provided with the radiative transfer package libRadtran. However, we are aware that such an approach could lead to significant uncertainties in the case of solar irradiances. We included the following lines in the text: "Standard aerosol profiles were used following the model by Shettle (1989). The aerosol type below 2 km was set to maritime for all the cases, while the season was modified according to the campaign periods."

ll. 124ff The authors should explain here how they define "cloudy", perhaps foreshadowing the use of ground-based net radiation that is currently mentioned much later. It would be important to translate the net radiation threshold into an optical thickness threshold for reference.

We included the following text to provide context about the two states. "Surface measurements typically indicate two radiative modes: a cloudless state with strongly negative TIR net irradiances, and a cloudy state with TIR net irradiances close to zero (Stramler et al., 2011)." Defining a threshold for cloud optical thickness is challenging as clouds in different altitudes have different effects on the net irradiance profiles. Cloud base temperature is often more important. However, for low-level clouds, we estimated that clouds above $\tau > 1$ will significantly change the $F_{net}$ profile to be classified as cloudy, and we indicated this value in the revised text.

Fig. 3 Please show the -30 W m$^{-2}$ as vertical (thin, gray) line in each panel.

The line was included. The new element was introduced in the caption: "The separation between cloudless and cloudy cases is indicated by a grey dotted line."

Fig. 5 Please include the interquartile range of respective profiles, perhaps as shading.

The percentiles are now shown in the figure. New text was added to the caption: "Average values are displayed as solid lines, and the values between the 25th and the 75th percentile are represented by a colored area."

Fig. 6 Please increase the vertical extent of this figure to better display fine changes.

The figure is now presented vertically.

ll. 233 ff Please explain whether the adiabatic assumption is realistic for Arctic clouds.

We are aware that Arctic clouds are not necessarily adiabatic. In general, the single-layer boundary layer clouds follow the adiabatic profile (Lawson et al., 2001; Klingebiel et al., 2015). As no in situ measurements of cloud microphysical properties are available, the adiabatic assumption is the best we have to construct a cloud. This approach is also applied by Cloudnet providing the LWC profiles. We changed the text to: "The corresponding profiles of liquid water content (LWC) were derived by vertically distributing the liquid water path maintaining the adiabatic LWC assumption used in Cloudnet."

Section 4: Satellite imagery of these cases could be helpful.

We did not address this point as the section was removed.

Fig. 7 and 9: Please add a line marking the altitude of 0 and also -30 degree Celsius to illustrate where there are supercooled and homogeneous freezing conditions.

We did not address this point as the figures were removed.

l. 273 The sentence suggests that the timeline shows the same airmass. Is that really the case or was there advection?

Advection is a possible explanation for this variation. However, now the section does not investigate the cause-effect relation but rather looks at the radiative effect of the cloud.

ll. 301-303 Would the inclusion of a high-altitude ice cloud in radiative transfer simulations produce plausible heating rate profiles? Perhaps include a simulation with the best guess in ice cloud properties. With respect to Section 5, the authors could use it an example of a relatively thin cloud where moderate boundary layer cloud-top cooling is permitted.

The main purpose of Section 5.2 was to show the effect of an elevated ice cloud on two different boundary layers: a cloudless one and one with a low-level liquid-bearing cloud. More specifically, the cases we use are the ones described in Section 4.1. As the reviewer suggests, including the ice cloud improves the simulations. We clarified the link between the two (revised) sections by adding: "This effect was analyzed in Sect. 4.2, and showed that adding the missing cloud results in a better match between simulations and observations."

ll. 326-331 Perhaps best reserve this for a discussion (see major point). Similar to an above concern, was there advection happening? If so, please rephrase to omit the misleading impression of having sampled the same airmass twice.

We did not address this point as the section was removed.

---

## Author Response (AR2)

**R1**

I liked the change in this version of the paper to include the discussion of cloud radiative effect in the new section 5. My only remaining comment is that this discussion is not well referenced in the conclusion section. There is a very brief discussion of the 30 September case discussed in section 5, but the cloud radiative effect findings are not discussed. Otherwise, I appreciated the changes in this version and have no further suggestions.

We thank the reviewer for the positive feedback. The text in conclusion section was integrated as follows:

"A profile of CRE was derived from the transition between atmospheric states observed with a sequence of balloon profile measurements obtained on 30 September 2021. A low-level cloud dissipated, and the vertical profiles of the radiative energy budget and the radiative cooling rates changed significantly and rapidly. The CRE from surface measurements matched the results obtained at the base of the balloon profile, however the CRE steadily increased with altitude up to cloud base height and peaked in the central part of the cloud. Simulations alternating the two thermodynamic profiles were combined with switching the cloud layer on and off, indicating that cloudiness is the main driver in controlling the structure of the TIR radiation profile."